# Application of Real-Time PCR Assays for the Diagnosis of Histoplasmosis in Human FFPE Tissues Using Three Molecular Targets

**DOI:** 10.3390/jof9070700

**Published:** 2023-06-25

**Authors:** Luisa F. López, Ángela M. Tobón, Diego H. Cáceres, Tom Chiller, Anastasia P. Litvintseva, Lalitha Gade, Ángel González, Beatriz L. Gómez

**Affiliations:** 1Medical and Experimental Mycology Group, Corporación para Investigaciones Biológicas (CIB), Medellín 050034, Colombia; lufe114@yahoo.com.mx (L.F.L.); diegocaceres84@gmail.com (D.H.C.); 2Instituto Colombiano de Medicina Tropical, Universidad CES, Medellín 055450, Colombia; angelamtobon@hotmail.com; 3Mycotic Diseases Branch, Centers for Disease Control and Prevention, Atlanta, GA 30333, USA; tnc3@cdc.gov (T.C.); frq8@cdc.gov (A.P.L.); hvr0@cdc.gov (L.G.); 4Basic and Applied Microbiology Research Group (MICROBA), School of Microbiology, Universidad de Antioquia, Medellín 050010, Colombia; angel.gonzalez@udea.edu.co; 5Studies in Translational Microbiology and Emerging Diseases (MICROS) Research Group, School of Medicine and Health Sciences, Universidad del Rosario, Bogota 111221, Colombia

**Keywords:** histoplasmosis, *Histoplasma capsulatum*, diagnosis, real time PCR, FFPE tissues

## Abstract

Histoplasmosis is a fungal infection caused by the thermally dimorphic fungus *Histoplasma capsulatum.* This infection causes significant morbidity and mortality in people living with HIV/AIDS, especially in countries with limited resources. Currently used diagnostic tests rely on culture and serology but with some limitations. No molecular assays are commercially available and the results from different reports have been variable. We aimed to evaluate quantitative real-time PCR (qPCR) targeting three protein-coding genes of *Histoplasma capsulatum* (100-kDa, H and M antigens) for detection of this fungus in formalin-fixed paraffin-embedded (FFPE) samples from patients with proven histoplasmosis. The sensitivity of 100-kDa, H and M qPCR assays were 93.9%, 91% and 57%, respectively. The specificity of 100-kDa qPCR was 93% when compared against samples from patients with other mycoses and other infections, and 100% when samples from patients with non-infectious diseases were used as controls. Our findings demonstrate that real-time PCR assays targeting 100-kDa and H antigen showed the most reliable results and can be successfully used for diagnosing this mycosis when testing FFPE samples.

## 1. Introduction

Histoplasmosis is caused by the thermally dimorphic fungus *Histoplasma capsulatum* and is one of the most common fungal infections. This pathogen lives in the environment and has been mostly found in soil with bat or bird droppings. The infection occurs when the infective particles of the fungus, microconidia and small fragments of hyphae are inhaled by an individual affecting primarily the lungs [1,2,3]. As the severity and dissemination of the infection depends mainly on the size of the inoculum and the immune system of the individuals, this mycosis causes significant morbidity and mortality in people who have weakened immune systems such as those living with HIV/AIDS, especially in countries that have limited access to rapid diagnostics or antiretroviral therapies, in which up to 48% mortality has been reported [1,2,3,4]. Histoplasmosis is most frequently diagnosed on the American continent and is often underdiagnosed in countries where the disease is highly endemic [3]. Currently, the diagnosis relies on conventional blood cultures, which are positive in approximately 50% of cases and may take up to 6 weeks to grow, thus delaying diagnosis and initiation of therapy [5]. Immunological tests that include specific antibody detection have showed low sensitivity because reported results may often be negative in immunocompromised patients [6]. Detection of circulating *Histoplasma* antigens in urine specimens is highly sensitive, reaching up to 95% positivity in certain patient populations [7,8]. In addition to acute pulmonary and disseminated disease, subclinical and asymptomatic infections that manifest as pulmonary nodules and mediastinal granulomas occur in immunocompetent people living in endemic areas. These infections are often accidently discovered during imaging and mistaken for malignancy; *H. capsulatum* is then presumptively diagnosed by histopathology on biopsy samples [9]. To achieve the definitive diagnosis of *H. capsulatum* in these samples, molecular confirmation would be especially beneficial to rule out cancer and other pathologies; however, DNA-based diagnostic tools are not routinely used for diagnosis of histoplasmosis [3,10]. PCR assays are not commercially available and only a few in-house tests amplifying different targets have been reported in the literature, showing variable results [11,12,13,14,15,16,17,18,19,20,21,22,23].

We recently standardized and validated a real-time PCR assay, using three protein-coding genes as molecular targets (100-kDa, M and H antigens) in an animal model of histoplasmosis using formalin-fixed paraffin-embedded (FFPE) and fresh lung tissues [24]. In the present study, we applied these real-time PCR protocols to detect *H. capsulatum* DNA in FFPE human samples.

## 2. Materials and Methods

**Formalin-fixed paraffin-embedded (FFPE) tissue samples.** Tissue samples were submerged and fixed in a solution containing 4% formalin, then embedded in paraffin and cut [25]. Ninety-seven archived FFPE tissue samples were studied. The samples were obtained from the Department of Pathology of the Hospital la María, Medellín, Colombia, between 2008 and 2011 from patients diagnosed by histopathological analysis with the following infectious diseases: histoplasmosis *n* = 36, tuberculosis *n* = 29, cryptococcosis *n* = 6, leishmaniasis *n* = 5, *Malassezia* infection *n* = 2, sporotrichosis *n* = 1, pneumocystosis *n* = 1, mucormycosis *n* = 1, cytomegalovirus *n* = 1, herpes virus *n* = 1, tinea infection *n* = 1, as well as 13 samples from patients with non-infectious diseases (prostatic hyperplasia *n* = 2, cholestasis *n* = 3, endometriosis *n* = 1, appendicitis *n* = 3, uterine prolapse *n* = 1, suprarenal hyperplasia *n* = 1, abdominal carcinoma *n* = 1 and soft tissue mass *n* = 1). FFPE tissues from different organs were processed for histopathological analysis and DNA extraction.

**Histopathological analysis.** FFPE tissue sections were stained with Grocott’s methenamine silver to identify *H. capsulatum* yeast cells. In addition, from each tissue block 5 cm sections were cut and used for DNA extraction as previously described [25].

**DNA extraction.** Nucleic acids were extracted using a QIAamp^®^ DNA FFPE Tissue Kit (Qiagen; Valencia, CA, USA) according to the manufacturer’s instructions and modifications previously described by Muñoz-Cadavid et al. [25].

**Real-time PCR amplification.** Three different protein-coding genes of *H. capsulatum* (100-kDa, H and M antigens) were selected and amplified in clinical samples as described previously. Primers, probes and protocols were used according to López et al. [24]. Each assay was tested in duplicate, and positive (*H. capsulatum* DNA) and negative controls (sterile distilled water) were included. Conventional PCR targeting the human β-globin housekeeping gene was performed in order to validate the presence of amplifiable DNA and absence of inhibitory substances, following the protocol described by Bialek et al. [26]. Additionally, human DNA (Promega, Madison, WI, USA) was used.

**Statistical analysis.** All analyses were performed using the software STATA version 8.0 and Epidat version 3.1. All data were saved in an electronic Microsoft Access^®^ database. Sensitivity, specificity, positive and negative predictive values were calculated using contingency tables with their respective 95% confidence intervals (95% CI).

## 3. Results

FFPE samples from patients previously diagnosed with histoplasmosis and other infectious diseases were processed and analyzed. Yeast cells were observed in 36 samples from patients previously diagnosed with histoplasmosis. From the 36 FFPE samples with visible Histoplasma-like yeasts, three did not amplify the β-globin gene and were excluded from the subsequent analysis. Thirty-one (93.9%) of the analyzed 33 samples (Table 1) with proven histoplasmosis showed amplification with at least one qPCR protocol: 19 (57.5%) FFPE samples were positive for the three protocols, 11 (33.3%) were positive for 100-kDa and H antigen and one (3%) was positive for 100-kDa only. Sensitivities were 93%, 91% and 57% for 100-kDa, H antigen and M antigen, respectively. DNA amplifications were performed from different tissue samples including lymph node (*n* = 19), skin (*n* = 11), liver (*n* = 1), pharynx (*n* = 1), and lung (*n* = 1) (Table 1).

A total of 29 FFPE samples from patients with other infectious diseases were analyzed using the three qPCR protocols to assess specificity, the remain 19 samples were excluded from the subsequent analysis since the β-globin gene did not amplify. Two (6.9%) samples from patients previously diagnosed with tuberculosis were found positive for histoplasmosis with the 100-kDa protocol, and the remaining 27 samples were negative with all three assays. The overall specificity for the molecular test using this group was 93% (95% CI) for the 100-kDa protocol and 100% for the H and M protocols. Total agreement was observed across the replicates for all the DNAs tested. In addition, no amplification with Histoplasma-specific primers was observed from 10 control FFPE samples from patients without histoplasmosis and any other infectious diseases. However, the human β-globin gene was successfully amplified from these 10 samples.

## 4. Discussion

In recent years, several diagnostic assays have been developed for acute pulmonary or disseminated histoplasmosis, which have considerably improved the diagnostic landscape for this infection; however, the need for rapid detection of *H. capsulatum* in tissues remains. Histoplasmosis is frequently discovered during histopathological examinations of biopsy specimens collected for other diagnostic purposes. Current diagnostic assays used for confirming the histopathology finding rely on PCR amplification and sequencing of rDNA genes, which is lengthy and labor-intensive. The development of a rapid molecular assay can considerably improve the diagnostics of histoplasmosis from tissues. Here, we described the performance of real-time PCR assays targeting 100-kDa protein, H and M antigen genes for detection of *H. capsulatum* in FFPE specimens from patients with histoplasmosis. Previously, we evaluated the performance of these assays using fresh and FFPE tissues from the murine model of histoplasmosis and showed 100% sensitivity of these assays one-week post-infection. Here, we evaluated the performance of these assays using clinical samples from patients to evaluate the utility of these assays for molecular diagnostics. 

The 100-kDa target has been used by others in the conventional, nested, and quantitative PCR assays for detection of *H. capsulatum* DNA with variable performance characteristics [12,14,15,16,17,27,28]. In this study, the assay targeting 100-kDa protein gene demonstrated 94% sensitivity using FFPE tissues from patients with confirmed histoplasmosis and 100% specificity using FFPE tissues from patients without infectious diseases. However, 94% specificity was observed when using FFPE samples from patients with other infectious diseases, as specimens from two individuals with tuberculosis tested positive. Coinfections between tuberculosis and histoplasmosis are common in patients with HIV, and sensitivity of microcopy is low, especially in individuals with low fungal burden [29,30,31,32]. For example, 27–33% of patients with HIV and progressive disseminated histoplasmosis in Guatemala were infected with tuberculosis [33]. Similarly, in Colombia, 35% of patients with histoplasmosis presented tuberculosis as co-infection [34]. Therefore, it is possible that the two control samples that generated positive results were also co-infected with *H. capsulatum*. 

The H and M antigen genes have not been previously used as targets for diagnostic qPCR assays in humans. Non-human samples have been used to validate a conventional PCR targeting M antigen in vitro, where a sensitivity of 100% was observed using DNA extracted from isolates [18]. A semi-nested PCR assay for the H antigen target was evaluated on human samples with 100% sensitivity and 92% specificity [19]. More recently, our group reported the successful use of these targets in an animal model showing 100% sensitivity and specificity of these assays when using tissues collected one week post-infection, although the sensitivities of both assays declined in the following weeks [24]. 

In this study, the performance of the H antigen assay was similar to that of 100-kDa assay: 91% sensitivity was observed when used FFPE tissues from patients with confirmed histoplasmosis, with only one sample positive by 100-kDa assay testing negative with H antigen assay. Furthermore, the specificity of H antigen assays was 100%: tissues from two patients with tuberculosis that tested positive by 100-kDa assay tested negative with H antigen suggesting that the specificity of 100-KDa primers and probes need to be further evaluated. Conversely, the performance of M antigen assay in FFPE tissues from patients with histoplasmosis was lower than the other two assay: several samples positive by both 100-kDa an H antigen assays testing negative with the M antigen assay for the overall sensitivity of 57%. 

Formalin fixation is known to interfere with PCR amplification. To validate the presence of amplifiable DNA, we used human β-globin housekeeping gene as a control and found no amplification in 26% of tested FFPE samples, which were excluded from the analysis. Studies have reported between 20% and 62% of inhibition [25,26], using the same target, and in addition, other authors reported 38% of inhibition using human glyceraldehyde-3-phosphate dehydrogenase (GAPDH) gene as target [12]. Our results are therefore in concordance with such findings and confirm the occasional effect of formalin fixation on DNA amplification, a process in which many factors are involved [29]. 

## 5. Conclusions

Overall, our results demonstrate that 100-kDa and H antigen assays show high sensitivity and specificity and the potential for further development into diagnostic assays. These assays may be especially useful for rapid diagnostic of *H. capsulatum* in biopsy tissues. Further evaluations are needed to test whether these assays may be useful for diagnosing indeterminate pulmonary nodules to rule out cancer. 

## Figures and Tables

**Table 1 jof-09-00700-t001:** FFPE tissue samples from patients with histoplasmosis: results of histopathological analysis and qPCR.

Sample Information		Histopathological Test *	Molecular Tests		
Patient ID	Tissue	Grocott’s Methenamine Silver Stain	100-kDa Protocol	H antigen Protocol	M antigen Protocol
1	Skin	+++	Positive	Positive	Positive
2	Skin	+++	Positive	Positive	Negative
3	Lymph node	+	Positive	Positive	Negative
4	Lymph node	+	Positive	Positive	Negative
5	Lymph node	+++	Positive	Positive	Positive
6	Lymph node	++	Positive	Positive	Negative
7	Lymph node	+++	Positive	Positive	Positive
8	Lymph node	+++	Positive	Positive	Positive
9	Skin	+++	Positive	Positive	Positive
10	Skin	++	Positive	Positive	Positive
11	Skin	+++	Positive	Positive	Positive
12	Lymph node	+++	Positive	Positive	Positive
13	Lymph node	++	Positive	Positive	Negative
14	Lymph node	++	Positive	Positive	Positive
15	Lymph node	+	Positive	Positive	Positive
16	Lymph node	+	Positive	Negative	Negative
17	Skin	+++	Positive	Positive	Positive
18	Lymph node	+++	Positive	Positive	Negative
19	Skin	+++	Positive	Positive	Positive
20	Lymph node	+++	Positive	Positive	Positive
21	Lymph node	++	Positive	Positive	Positive
22	Skin	++	Positive	Positive	Positive
23	Liver	++	Positive	Positive	Positive
24	Lymph node	+++	Positive	Positive	Positive
25	Lymph node	++	Positive	Positive	Negative
26	Skin	+++	Positive	Positive	Positive
27	Lymph node	+++	Positive	Positive	Positive
28	Skin	++	Positive	Positive	Negative
29	Skin	+++	Positive	Positive	Negative
30	Lymph node	++	Positive	Positive	Negative
31	Pharynx	+++	Positive	Positive	Negative
32	Lymph node	++	Negative	Negative	Negative
33	Lung	+	Negative	Negative	Negative

* Evaluation of the FFPE tissue sections for the presence of the characteristic yeast-like cells in high (+++) or low (+ to ++) numbers.

## Data Availability

Not applicable.

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
