# Peer review of "Application of Real-Time PCR Assays for the Diagnosis of Histoplasmosis in Human FFPE Tissues Using Three Molecular Targets"

_jof, 2023, doi:10.3390/jof9070700_

Round 1
Reviewer 1 Report
This is an interesting study on application of rtPCT for detection of 3 Histoplasma genes in FFPE samples.
The authors report very high sensitivity and specificity of 2 of 3 targets.
I would like the authors to
1) comment if the type of sample (which wer mainly lymph nodes) cold impact the rat of positivity
2) comment if Grocotts' staining revision was performed for PCR samples from TB patietns
Author Response
1) comment if the type of sample (which were mainly lymph nodes) could impact the rate of positivity
R/ Thank you for this comment. We do not believe that lymph nodes impact the rate of positivity because all the samples were from patients previously diagnosed with histoplasmosis and additionally, all the samples including skin, lung, and liver were also positive for Histoplasma capsulatum when silver stain was analyzed. Although we think that the type of sample could impact the positivity of the technique, more different types of samples, including fresh samples, should be tested.
2) comment if Grocotts' staining revision was performed for PCR samples from TB patients
R/ Thanks for this important question; however, we did not perform Grocotts’ staining analysis for samples from patients diagnosed with TB. Although we are aware that this co-infection could be approximately 20%, the qPCR is quite specific. Moreover, our main objective was to amplify Histoplasma DNA.
Reviewer 2 Report
Detection of Histoplasma antigen in blood or urine is a sensitive method for rapid diagnosis of disseminated and acute pulmonary histoplasmosis, but is insensitive for chronic forms. In non endemic areas is more frequent the differential diagnosis from micobacteriosis or malignancy. Histoplasmosis is often accidentally income. A molecular test that help in diagnostics where there are technical difficulties like in paraffinated material is very interesting. Each assay was tested in duplicate, it would be interesting to know if there was agreement in the repetitions. The reference are appropriate, the conclusions are sufficient to explain the study results. The table is clear."Overall, our results demonstrate that 100-kDa and H antigen assays show high sensitivity and specificity and the potential for further development into diagnostic assays. These assays may be especially useful for rapid diagnostic of H. capsulatum in biopsy tissues. Further evaluations are needed to test whether these assays may be useful for diagnosing indeterminate pulmonary nodules to rule out cancer" this paragrph of Discussion could be included in the Conclusions.
97 FFPE samples are cited in materials and methods for the study but 72 samples are reported in the results: the other 25?
Author Response
1. Each assay was tested in duplicate, it would be interesting to know if there was agreement in the repetitions. The references are appropriate, the conclusions are sufficient to explain the study results. The table is clear.
R/ Thank you for the comment. The agreement in the repetitions was 100% and was described in lines 118 and 119.
- "Overall, our results demonstrate that 100-kDa and H antigen assays show high sensitivity and specificity and the potential for further development into diagnostic assays. These assays may be especially useful for rapid diagnostic of H. capsulatum in biopsy tissues. Further evaluations are needed to test whether these assays may be useful for diagnosing indeterminate pulmonary nodules to rule out cancer" this paragraph of Discussion could be included in the Conclusions.
R/ Thanks for the comment. The paragraph was moved to the Conclusions as suggested.
- 97 FFPE samples are cited in materials and methods for the study but 72 samples are reported in the results: the other 25?
R/ Samples with positive Grocott’ staining for Histoplasma-like yeasts (n=36) are described in lines 100-101; however, as described in M&M, “three did not amplify the β-globin gene and were excluded from the subsequent analysis”.
For other infectious diseases (n=48), 19 were excluded because the human β-globin gene did not amplify. This clarification was included in the new version in lines 113 and 114.
Samples from patients without histoplasmosis and any other infectious diseases (n=13): in line 122 we added the number “10” for clarification “the human β-globin gene was successfully amplified from these 10 samples”.
In summary, three positive samples for Grocott staining, 19 samples for other infectious diseases and three samples from patients without histoplasmosis or any other infectious diseases were excluded because the human β-globin gene did not amplify (n=25).